# Foreign Healthcare Professionals in Germany: A Questionnaire Survey Evaluating Discrimination Experiences and Equal Treatment at Two Large University Hospitals

**DOI:** 10.3390/healthcare10122339

**Published:** 2022-11-22

**Authors:** Elif Can, Clara Milena Konrad, Sidra Khan-Gökkaya, Isabel Molwitz, Jawed Nawabi, Jin Yamamura, Bernd Hamm, Sarah Keller

**Affiliations:** 1Department of Radiology, Berlin Institute of Health, Charité—Universitaetsmedizin Berlin Corporate Member of Freie Universitaet Berlin, Humboldt-Universitaet zu Berlin, Charitéplatz 1, 10117 Berlin, Germany; 2Department of Patient and Care Management, Integration and Anti-Racism Commissioner, University Medical Center Hamburg-Eppendorf (UKE), Martinistraße 52, 20246 Hamburg, Germany; 3Department of Diagnostic and Interventional Radiology and Nuclear Medicine, University Medical Center Hamburg-Eppendorf (UKE), Martinistraße 52, 20246 Hamburg, Germany; 4Evidia Group, Alice-Salomon-Platz 2, 12627 Berlin, Germany

**Keywords:** discrimination, diversity, migrant health workers, quantitative survey, workplace integration

## Abstract

Objective: To identify facilitators and barriers and derive concrete measures towards better workplace integration of migrants working in the German healthcare sector. Design: Two-centre cross-sectional quantitative online survey of experiences of discrimination among healthcare professionals with a migration history in two large German university hospitals. Participants: 251 participants fully completed the questionnaires. Main outcome measures: Experiences of discrimination and perception of inequality. Results: Fifty-five percent of migrant health workers had had at least some command of German before arriving in Germany. Members of all professional groups surveyed expressed experiences of discrimination related to language, nationality, race/ethnicity, and sex/gender. The proportions of staff with experiences of discrimination by peers differed significantly among occupational roles, with nurses and technologists having the most experiences of discrimination. The perception of inequality was reported more frequently than experiences of discrimination and had a negative impact on workplace satisfaction. Specifically, the compulsion to compete was a frequent feeling stated by participants. Conclusion: The mechanisms of discrimination and structural inequality revealed by our survey could inform specific measures, for example at the management level, to increase workplace satisfaction and attract migrant health workers in the long term.

## 1. Introduction

There is a demand for medical professionals throughout Europe that cannot be met locally or nationally. Globally, people from all medical disciplines are being recruited to work in different medical institutions and at all levels of hierarchy and responsibility in other countries [1,2].

In Germany, roughly a quarter of the population, or 21.2 million people, have a so called ‘migration background’ [1]—a concept that figured prominently in the discourse of the early 2000s. It widened the scientific and public understanding of migration-related, intergenerational transmissions of inequalities, regardless of citizenship, and contributed to the recognition of German society as one structured and shaped by migration. However, it also had exclusionary effects, especially due to the way the Federal Statistical Office operationalized the official concept of “migration background” and represented the microcensus results [1]. The label “migration background” is misleading, since only certain migration experiences are considered in the definition and handed on to descendants, while others are not. A “migration background” is ascribed to grandchildren of people born as foreigners, but not to children of immigrated German-born persons. The concept is grounded on citizenship, not migration experience. Given that the German citizenship law was historically based on blood ties, it is still an “ethnic” rather than a migration category [3]. While systematic data on the migration status of medical professionals in Germany are not available, recent estimates of the German Medical Association suggest that, of the approximately 392,400 doctors, almost 49,000 are foreigners, and the federal government estimates that, of the 1.7 million nurses working in Germany, about 133,900 have a foreign background [1,4].

The COVID-19 pandemic has led to a further increase in the demand for qualified staff for the medical sector, and global recruitment activities have also expanded [5,6].

Migration to countries belonging to the global North has become a fact and is motivated by inequalities at different scales, including economic development, income, democracy, and freedom [7,8,9]. Although written within the context of Italy’s period of industrialization, Gramsci’s “southern question” holds value to social science inquiries regarding relationships between the global North and the global South in terms of production and exploitation [10,11]. While the central promises of modern democracies attract migrants, there is growing support for right-wing populist positions, and discrimination continues to be a relevant social problem or may even be on the rise again [7,12,13,14]. 

Experiences of discrimination have long been a central part of social discourse. In addition to the still virulent—but at least publicly ostracised—biologist racism, in which alleged human characteristics are incorporated and regarded as hereditary, a trend towards the culturalization of racism can be consistently observed in international research in recent decades [15,16,17,18].

The characteristics of social groups are naturalised and narratively embedded based on central signifiers such as “culture”, “ethnicity”, “religion”, or “nationality” to legitimise social ostracism or social inequalities based on the perceived deviation of the culturalized or racialized groups [12]. While the German federal law called “Allgemeines Gleichbehandlungsgesetz”, aimed at preventing and eliminating discrimination in the workplace based on race, ethnic origin, gender, religion or belief, disability, age, or sexual identity, provides the legal framework for equal treatment, persons with a migration background still struggle for recognition, participation and equality and against discrimination, highlighting that the legal framework alone is not sufficient to ensure equal treatment for all people.

Discrimination towards employees in the healthcare sector manifests itself in interactions between employees and colleagues, employees and superiors, and employees and patients.

A few semi-quantitative and qualitative surveys on this topic have also been conducted in Germany. In her recent medical thesis, Klingler [19] conducted 20 semi-structured interviews with foreign-born and foreign-trained physicians working in German hospitals. Most struggled with their lack of facility-specific (linguistic, cultural, clinical, and systemic) knowledge. In addition, the behaviour of patients and colleagues/superiors was perceived as discriminatory or inappropriate for other reasons.

In a multi-centre cross-sectional study conducted among internationally trained nurses (*n* = 64) and host nurses (*n* = 103) at two university hospitals in Germany between August 2019 and April 2020, Roth et al. [20] reported that nurses who migrated to Germany mainly expected better working conditions, a higher standard of living and professional development opportunities. Their observations suggest that the expectations migrant nurses had before migrating may not be met, which in turn could have a negative impact on integration and their willingness to stay. With the increasing recruitment of internationally educated nurses from not only from other European countries, but also from overseas, it is crucial to identify factors that retain migrant nurses and promote their integration.

Several other European studies focused on the labour market integration of medical healthcare professionals in terms of language acquisition, qualification recognition, and onboarding processes [17,18,19,20,21,22,23,24,25,26].

Working conditions and job satisfaction are essential motivators for individuals and their quality of life in general. Measures to overcome current problems are thus also worthwhile for medical facilities wishing to attract qualified staff from other countries to meet their human resources needs. However, to date, potential barriers for and discrimination against migrant health professionals exist, or are subjectively perceived to exist, and may affect their general satisfaction and their will to stay permanently in Germany. The two studies cited above were conducted either among nurses only (Roth et al.) or among physicians (Klingler). Moreover, they were not quantitative surveys.

We conducted the first two-centre quantitative analysis on experiences of discrimination among healthcare staff who migrated to Germany.

The aim of this survey was to identify facilitators of and barriers to workplace integration from the health workers’ perspective that could inform concrete measures for improvement that could ultimate help in attracting and retaining urgently needed foreign staff.

## 2. Methods 

### 2.1. Study Design 

We conducted a cross-sectional questionnaire survey among the staff of two large German university hospitals: Charité–Universitätsmedizin Berlin (Charité) and Universitätsklinikum Eppendorf in Hamburg (UKE). A cross-sectional approach was chosen in order to make participation more attractive and to recruit enough subjects for quantitative data analysis.

To identify our target group, we used intelligent linkage to ensure that only responders not born in Germany could answer all of the questions. This was necessary because we initially had to send email invitations to participate in our survey to all employees of the two hospitals, since prior information on migrant status was not available or not accessible.

### 2.2. Questionnaire Survey

The study was conducted from October 2020 to June 2021.

The questionnaire was distributed online and was available to participants from October 2020 to February 2021 (Charité) and from March to June 2021 (UKE). A flowchart of participation and study inclusion at the two university centres is provided in Figure 1.

For correct extraction, we used decision questions. The questions presented in the online survey were logically linked in that the answer to one question determined which questions were asked next (intelligent linkage). 

### 2.3. Questionnaire

The questionnaire was developed using LimeSurvey (https://www.limesurvey.org/en/ accessed on 21 September 2022) and included a total of 97 questions. The questionnaire was developed with the support of the Institute for Biometry and with expert input in the definition of items from the Integration and Anti-Racism Commissioner of UKE. Participants could choose between an English and a German version (the English version of the questionnaire is provided as a Appendix A).

LimeSurvey guides access to questions based on how a respondent answers previous questions. The use of these intelligent linkages allowed personalisation of the questionnaire by decision questions, which minimised the number of questions for each responder. This approach was chosen in order not to deter potential participants by too many questions and explains the variation in sample sizes for different items of the questionnaire. The questions were arranged in seven subsections: demographics, profession and career, labour market access, working environment, language, private/social life, integration/support. Responsible at UKE requested some changes in the wording of the section on working environment (see the English version of the questionnaire in the Appendix A, where we have highlighted the differences).

Our questionnaire included different question types, such as multiple choice questions and questions requiring numerical or text input. A positive ethics vote and data protection support were given.

### 2.4. Statistical Analysis

Statistical analysis was performed in SPSS 27 Statistics for Mac OSX (SPSS Inc., Chicago, IL, USA). Data are presented using descriptive statistics including mean, standard deviation (SD), minimum (Min), maximum (Max) and percentiles (25, 50 and 75) as metric variables. Categorical data are presented as absolute numbers (*n*) and percentages. The independent-samples test was performed to compare the means of specific groups. The χ^2^ test was used for comparing groups, and *p*-values < 0.05 were considered to indicate statistically significant differences. 

## 3. Results

### 3.1. Study Population

Fully completed questionnaires were available from 142 responders at Charité and 109 responders at UKE (see the flowchart in Figure 1). The average completion time was 35 min. The questionnaire was completed in English by 28% (31/142) and 48% (52/109) of responders, respectively. The participants included in our analysis had a median age of 36 ± 8 years (min 22|max 59) at Charité and 35 ± 8 years (min 22|max 60) at UKE. Additional demographic information is presented in Table 1.

To identify possible differences between professional groups, we subdivided participants into three categories: clinical and scientific staff, junior staff, and non-scientific staff. The latter category included nurses and technologists such as medical or radiologic technologists. The distribution of categories and positions is summarized in Table 2.

About 53% (Charité) and 69% (UKE) of participants had already completed their professional training before coming to Germany.

At Charité 58.5% (83/142) females and 37.3% (53/142) males completed the questionnaire versus 66.1% (72/109) and 31.2% (34/109) at UKE. Females accounted for roughly three-quarters of nurses and technologists (73.1% (104/142) and 74.5% (81/109), respectively). In the category of clinicians and scientists, 51.2% (72/142) and 59.6% (65/109), respectively, were female.

Regarding countries of origin, 33.1% (40/142) of the respondents at Charité and 30.3% (33/109) at UKE were from EU countries. Details on countries of origin are provided in Figure 2 and Table 3.

### 3.2. Labour Market Access

The range between arrival in Germany and beginning of employment was six years at Charité and four years at UKE.

To evaluate access to the labour market, the recognition processes for professional degrees and qualifications, as well as administrative difficulties, were evaluated.

Responders with degrees obtained abroad (Charité: 53% (75/142); UKE: 69% (75/109)) were asked about the recognition of their qualification and if they felt discriminated against in dealing with German authorities.

At Charité, 66% (47/71) and at UKE 78% (53/68) did not feel discriminated against in dealing with German authorities.

When asked about their current positions, only 12% (17/142) of the participants at Charité and 12% (13/109) at UKE were employed in leadership positions.

The majority of the respondents felt that their current positions were adequate for their qualifications (Charité 78% (108/137; 5 were not available (NA), UKE 72% (76/106; 3 = NA)), while 18% (26/142) at Charité and 25% (27/109) at UKE did not. Altogether, only 2% at Charité and 1% at UKE stated that they were employed above their qualifications (Figure 3). 

### 3.3. Language

Participants had a wide range of native languages; 57% at Charité (79/139) and 57% at UKE (61/107) had at least some command of German before arriving in Germany.

The professional groups with the highest patient contact and communication, namely nurses and technologists, followed by physicians (residents and specialists), reported the most proficient German language skills. Researchers tended to report poorer to no knowledge of German. Overall, reported language skills were significantly different between the three professional categories (*p* < 0.001) (Table 1; Figure 4).

At both hospitals, the most significant language difficulties were reported for the following activities: documentation (Charité 33%, UKE 34%), telephone conversations (Charité 29%, UKE 42%), communication with colleagues (Charité 26%, UKE 39%), conflictual discussions (Charité 21%, UKE 31%), and patient communication (Charité 10%, UKE 20%). The results for the language difficulties assessments are not shown in a table or figure.

However, experiences of discrimination at their workplace, which participants were asked to report for the past six months, were not related to language skills (*p* = 0.402) or language-associated difficulties in everyday work (*p* = 0.602).

The results for the language difficulties assessments are not shown.

### 3.4. Workplace Discrimination Experiences

We specifically asked about experiences of discrimination by superiors, peers, and patients. At Charité, precise factors for the experience of discrimination were then queried (Table 4).

About 10% (5/52) of non-scientific staff at Charité felt discriminated against by superiors (Table 4).

At UKE, where the questionnaire specifically asked about a migration background as a possible cause of discrimination, 22% (11/51) of nurses and technologists felt discriminated against by superiors because of their migration background (Table 5).

The proportions of the staff with experiences of discrimination by peers differed significantly among professional groups at Charité (*p* = 0.006) and UKE (*p* = 0.002), with nurses and technologists reporting the most experiences of discrimination.

Experiences of discrimination related to language, nationality, race/ethnicity, and sex/gender were evaluated across all professional groups. The survey asked about discrimination in professional and private life, including questions about well-being in the work environment. A total of 66.2% (94/142) of respondents at Charité and 57.8% (63/109) at UKE stated that they were satisfied or very satisfied with their work environment, while 22.5% (32/142) at Charité and 44% (48/109) at UKE stated that they were partly satisfied. On the other hand, 9.8% (14/142) at Charité and 6.4% (7/109) at UKE were dissatisfied with the work environment. Meanwhile, 83% of respondents (118/142 at Charité and 91/109 at UKE) felt respected by colleagues.

Of the clinical and scientific staff at Charité, 9% (4/46) indicated discrimination by superiors, giving as possible reasons nationality (75%), ethnicity (50%), and/or language (50%).

Of the junior staff at Charité, 11% (4/36) felt discriminated against; the main reasons were language (75%) and nationality (75%). Only 8% (3/38) of the junior scientific staff at UKE felt discriminated against by their superiors.

In contrast, experiences of discrimination by peers were reported by 31% (16/52) of the non-scientific staff at Charité.

The main reasons given were language (44%), nationality, race and ethnicity, religion and social class (25%), name (19%), and sex or gender (13%).

At UKE, 35% (18/51) of the non-scientific staff felt discriminated against by peers.

Of the clinical and scientific staff at Charité, 17% (8/46) said they felt discriminated against by peers. Language (50%), gender (37%), and/or nationality, race and ethnicity, physical appearance, and age (25%) were indicated as the main reasons.

A total of 6% (2/36) of the junior staff at Charité indicated discrimination by peers; all of them attributed it to gender and language (100%), and half of them also to race and ethnicity (50%). Nearly twice as many responders in the junior staff group at UKE 13% (5/38) felt discriminated against by peers. About 25% (13/52) of the non-scientific staff at Charité felt discriminated against by patients.

In terms of the type of discrimination, 46% indicated race and ethnicity, 38% nationality and/or physical appearance, 31% age and/or language, 23% name and/or religion and 15% gender as possible explanations.

At UKE, 18% (9/51) of the non-scientific staff felt discriminated against by patients.

About 13% (6/38) of the clinical and scientific staff at Charité stated that they had been discriminated against by patients, of whom 67% stated race and ethnicity and 33% age as the reasons for the discrimination.

Around 14% (2/14) of the clinical and scientific staff at UKE felt discriminated against by patients.

Specifically, a compulsion to compete more than others was a frequent feeling described by participants: 50% (71/142) of the respondents at Charité and 66% (72/109) at UKE stated that they felt they had to prove themselves to their colleagues.

### 3.5. Perception of Equality at Work

Most foreign-born employees perceived equal treatment concerning various aspects of their work, such as, for example, their contracts, weekly working hours, working times, payment and opportunities to express their opinions in discussions among colleagues.

However, at Charité 38.8% (79/129) of respondents and 33% (63/94) at UKE indicated unequal opportunities for further training and promotion. At both hospitals, a quarter of participants indicated unequal treatment regarding the appreciation of work performance by superiors. At UKE, a similar number of respondents denied being treated equally concerning the appreciation of work performance by colleagues and patients, as well as the distribution of tasks within their team (Table 6).

## 4. Discussion

The focus of our survey was on the respondents’ professional satisfaction and experiences of discrimination at work. Our questionnaire specifically addressed different aspects of discrimination and professional integration, which we analysed for different professional groups, including physicians and scientists, nurses and technologists, and junior staff such as students and postdocs.

(Labour) migration to post-national-socialist Germany led to diversification and demographic changes that altered Germany’s homogeneity, which had been violently manufactured by Nazi racial mania. Almost one-third of the total German population (26.7%) (1) has a migration history. In Germany, racism is closely linked to questions of migration and changing ideas of integration and is mainly directed against people who have come to Germany as migrants or refugees—or are perceived as such—as well as their descendants. A European comparison also shows that racism in Germany is primarily located in the context of migration and integration [26,27,28,29,30,31].

Concerning recent discrimination experiences at work, the questionnaire asked about discrimination by superiors, colleagues on the same hierarchical level, and patients. For use at UKE, this general question had to be modified and was specifically related to participants’ migration backgrounds as the underlying cause, while at Charité, the question was phrased more generally, and respondents could choose from a set of factors to which they attributed their experiences of discrimination (Questions 13–16 in the English version of the questionnaire provided as Appendix A).

Across all three professional groups we distinguished, most respondents did not feel discriminated against by their superiors. However, at both sites, more than a third of all nurses and technologists felt discriminated against by peers, and about a quarter by patients. Discriminatory behaviour was mainly caused by peers, colleagues from other professions and patients. In all cases, various forms of discrimination were reported, mainly based on language, nationality, race or ethnicity and gender.

Earlier studies [19,20,21,22,31,32,33,34,35,36,37,38,39,40,41] reported that international nurses felt distressed, confused, and humiliated by the discrimination they experienced while working. Participants reported incidents of discrimination including ‘patients displaying racist behaviour’, ‘refusal of care by international or Black nurses’ and ‘staff undermining the work of their international colleagues or drawing unfair conclusions about morale, motivation or character’ [26,35,36,37,38,39,40,41,42,43,44,45,46,47].

Likupe’s study [34] of Black African nurses working in the UK’s National Health Service (NHS) is of particular interest because it not only investigates experiences of racism and discrimination from the perspective of Black nurses, but also explores their managers’ perspectives on these issues. Black African nurses felt that their experience and knowledge of nursing was not respected. They considered it racist when they were ignored by patients and their relatives, giving them a feeling of being regarded as incompetent.

They also reported that they were prohibited from performing certain procedures and that the tasks they were assigned reflected a lack of trust in Black African nurses by their supervisors. Those findings are supported by various other studies [37,38,39,40,41,42,43,44,45,46]. The experiences of discrimination (except discrimination by patients) in our study population turned out to have a negative influence on self-confidence and workplace satisfaction of most responders and thus are in line with the studies just quoted. Perceptions of inequality were indicated more frequently than experiences of discrimination and showed a negative impact on workplace satisfaction as well.

Although, in our study, most respondents felt respected by superiors, colleagues, and patients with regard to their work performance, significantly more than half of all respondents at Charité and UKE stated that they felt they had to prove themselves to their colleagues. Depending on structural offers, discrimination becomes verbalizable in the first place. The same applies to the different assessments of individual discrimination and collective discrimination (person–group discrepancy). On the other hand, “perpetrators” can evaluate their own behaviours towards the victims as non-discriminatory and thus legitimate [31]. Sellers and Shelton [33] surveyed 267 African-American students regarding self-perceived discrimination processes. More than half of the respondents reported discriminatory experiences. The most frequent incidents related to brief encounters (being ignored, being overlooked, not receiving service, being treated rudely).

Thus, the feeling of having to prove oneself in front of colleagues shapes all aspects of discrimination [25,33,48].

In two of the three professional groups into which we divided the study participants—non-scientific staff and clinical and scientific staff—at Charité, language was selected as the main discriminatory factor by well over half of those who felt discriminated against. About a quarter in both categories cited nationality, race, and ethnicity as reasons. More than a third of all clinical and scientific staff ticked off gender as a discriminating factor, which clearly distinguishes their responses in this category from those of the technologists and nurses.

As most nurses are female, gender discrimination appears to be less of a problem in this category, while female doctors seem to be under greater competitive pressure from their male colleagues, to whom they are about equally distributed in percentage terms [33,38,48].

Another finding of our study was that respondents who had already acquired language skills prior to their immigration and considered their command of German to be rather good felt the least supported by their institutions in their efforts to further improve their German skills by attending job-specific language classes.

When asked what kind of support they thought would be most useful, participants particularly mentioned topics related to the period immediately after arrival in Germany, such as help with formalities or going to the authorities, which were mentioned most often, followed by education and housing and acquiring language skills. Support mechanisms for private commitments/life were also mentioned following the survey, including most frequently social life/events and mobility education and training, childcare, school and raising children.

The study of Likupe [34] also examined barriers to advancement for Black African nurses from the perspective of nursing managers. Some managers in the study admitted that they did not discuss professional development plans with Black African nurses. Nurses and managers reported that Black African nurses face racism, discrimination, and a lack of equal opportunities in the British National Health Service (NHS).

The mechanisms of discrimination and structural inequality revealed by our survey could also inform specific measures, for example at the management level, to increase workplace satisfaction and attract migrant health workers in the long-term.

Such an approach could also work in the German health system.

## 5. Conclusions

We here present the first survey of experiences of discrimination among all foreign-born staff, regardless of their qualifications and positions, at two large university hospitals in Germany.

The data generated in this cross-sectional study are exemplary for the university health sector in Germany and can inform future structural and personnel policies aimed at keeping the health sector attractive for foreign skilled workers in the future. Even if the respondents did not feel discriminated against primarily by their superiors, they are the ones who determine and can change structural work processes to a certain extent. Therefore, involving superiors in the process of awareness-raising may help bring about change and improvement.

In addition, larger-scale longitudinal quantitative surveys of the work situation in hospitals could be conducted in the future. Such studies might also address multiple causes of discrimination, as well as the possible effects of structural measures implemented to overcome discrimination in the workplace.

## 6. Limitations

The small sample size and low response rate are major limitations of our survey. In this respect, it must be considered that a large proportion of the employees contacted, as indicated in the method section, may not have received the request for participation due to absences, e.g., due to maternity/parental leave, extended illness, sabbaticals or similar. Unfortunately, it was not possible to specify the absences due to data protection considerations. Furthermore, there was no control group for the already small study population. The small sample size precluded subgroup analysis of discrimination mechanisms in relation to class, gender, and race. The questionnaires used at the two hospitals differed slightly in the phrasing of some questions. The questionnaire was only available in German and English. We have no information on the English proficiency of the participants who chose the English version. The possible influence of the COVID-19 pandemic as a confounder was not recorded.

## Figures and Tables

**Figure 1 healthcare-10-02339-f001:**
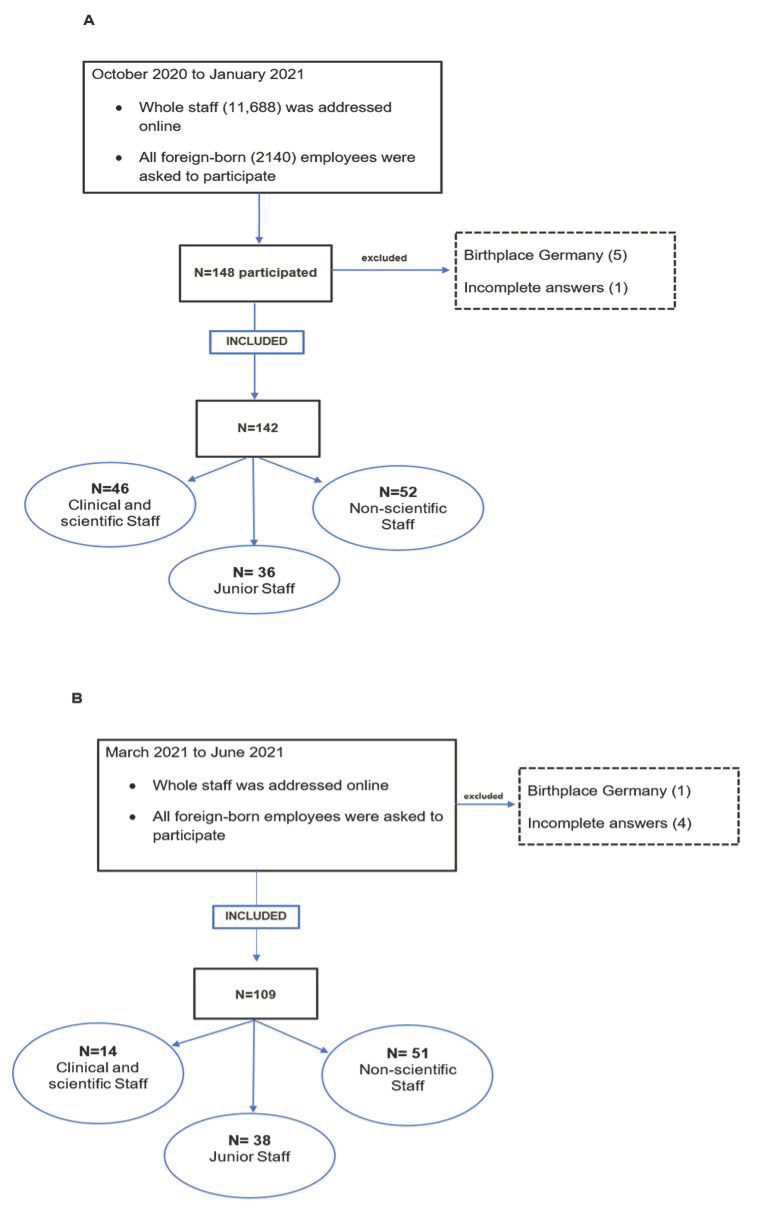
Flowchart of questionnaires included in the final analysis based on migrant status and exclusion at the two German university centres at which our survey was conducted. Information on the country of birth of all employees was available at Charité (**A**) but not at UKE (**B**). Abbreviations: Charité = Charité–Universitätsmedizin Berlin, UKE = Universitätsklinikum Eppendorf, Hamburg.

**Figure 2 healthcare-10-02339-f002:**
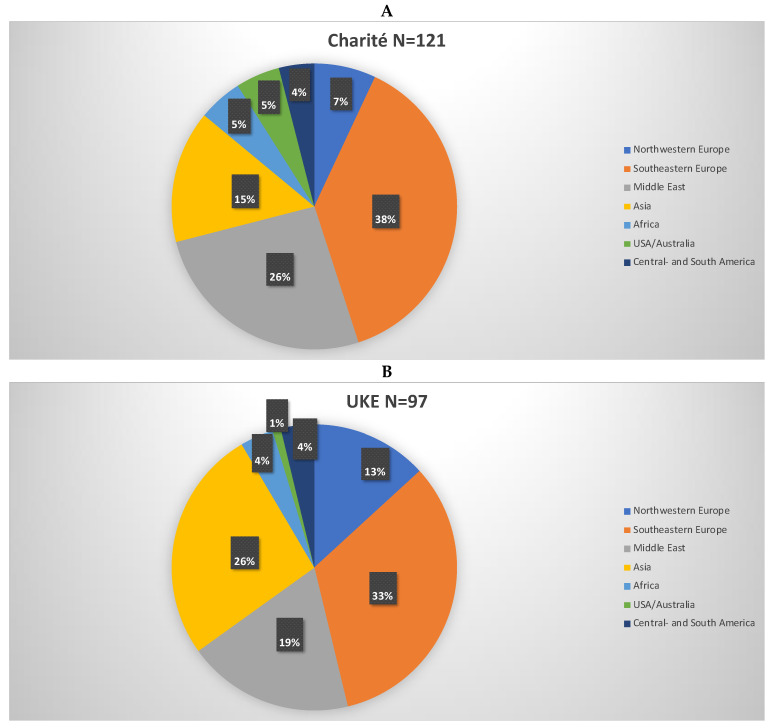
Distribution by EU and Non-EU countries of survey participants at Charité (**A**) and UKE (**B**). The pie chart shows how many different countries and language areas the participants originated from. EU nationals do not need visas/residence permits and have the equivalence of their qualifications automatically recognized.

**Figure 3 healthcare-10-02339-f003:**
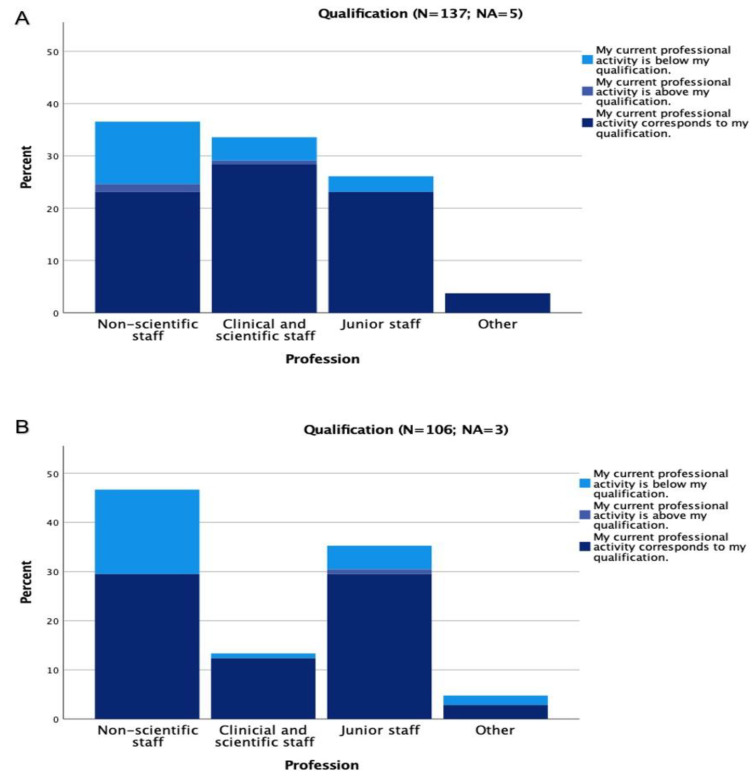
Current positions in relation to qualifications at Charité (**A**) and UKE (**B**). In order to assess access to the labour market, the recognition procedures for professional degrees and qualifications as well as administrative difficulties were evaluated. The majority of respondents felt that their current position was appropriate to their qualifications.

**Figure 4 healthcare-10-02339-f004:**
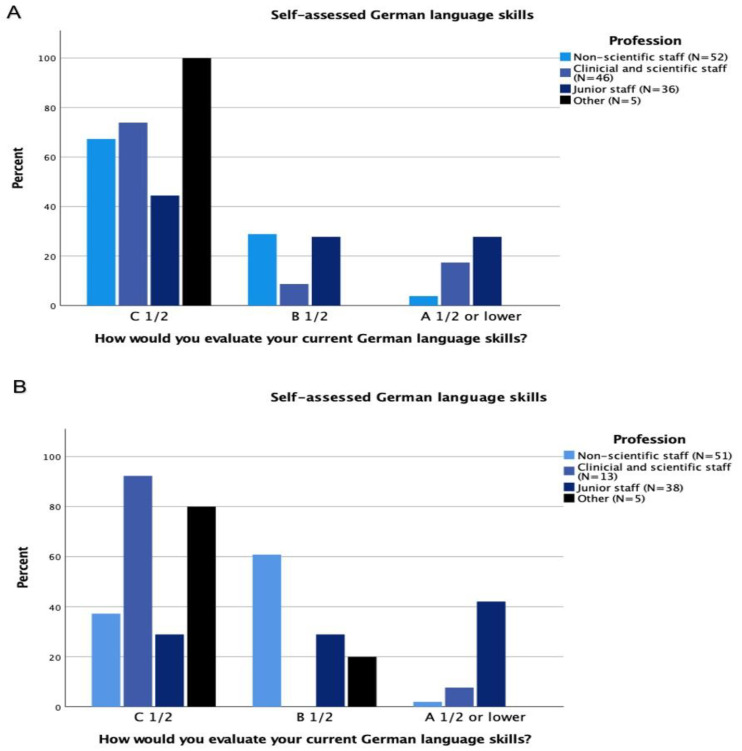
German language skills at Charité (**A**) and UKE (**B**) were assigned to one of three proficiency levels: C1/2—Advanced/Fluent; B1/2—intermediate; and A1/2—Beginner. The occupational groups with the most patient contact had the best command of German. Overall, the reported language skills differed significantly between the three occupational groups (*p* = 0.001), with clinical and scientific staff reporting the most proficient language skills.

**Table 1 healthcare-10-02339-t001:** Demographic Characteristics of Participants. The mean age of the participants (*n* = 132; NA = 10) was 36 years (SD: 8.834) at Charité and 35 years at UKE (SD: 7.711). At Charité, the mean duration of employment (*n* = 141; NA = 1) was 6 years (SD: 6.576), and the time between arrival in Germany and year of employment (*n* = 131; NA = 11) was 6 years (SD: 10.719). At UKE, the mean duration of employment (*n* = 104; NA = 5) was 4 years (SD: 4.735), and the mean time between arrival in Germany and year of employment (*n* = 101; NA = 8) was 4 years (SD: 7.456).

	Charité		UKE	
	*n*	%	*n*	%
Gender	136	100	107	100
Female	83	61	72	66
Male	53	39	34	34
Other	0		1	1
Country of birth	121	100	97	100
EU	40	33	33	30
Non-EU	81	67	64	59
Degrees	132	100	107	100
Abroad	75	53	75	69
Germany	57	40	29	27
Vocational training	25	17.6	15	14
University degree	110	77.5	92	86
Bachelor	25	17.6	36	33
Master	27	19	29	27
Diploma	7	5	8	7
PhD/doctorate	12	8.5	14	13
Medical Doctor	46	32		
Other	4	3	5	4.5
Citizenship	142	100	109	100
German	44	31	26	24
Non-German	98	69	83	76
Residence status	98	100	77	100
Temporary (only employees born outside the European Union)	41	43	38	49
Permanent	54	57	39	50
NA	3	3	6	8
Survey language	142	100	109	100
German	102	72	57	52
English	40	28	52	48

**Table 2 healthcare-10-02339-t002:** Surveyed professional categories and subgroups at each of the two participating sites. Individuals who did not provide information on their profession were recorded as not available (NA).

	Charité		UKE	
Position	N	Percentage (%)	N	Percentage (%)
Three categories:				
Clinical and scientific staff	46	32.4	14	12.8
Junior staff	36	25.4	38	34.9
Non-scientific staff	52	36.6	51	46.8
Other	5	3.5	5	4.6
NA	3	2.1	1	0.9
Total	142	100	109	100
Clinical and scientific staff				
Resident	22	47.8	4	28.5
Specialist doctor	10	21.7	6	42.9
Researcher	13	28.3	2	14.3
Other	1	2.2	0	0
NA	0	0	2	14.3
Total	46	100	14	100
Junior staff				
PhD student	14	38.9	19	50
Postdoc	13	36.1	18	47.4
Other	6	16.7	1	2.6
NA	3	8.3	0	0
Total	36	100	38	100
Non-scientific staff				
Nurse	39	75	43	84.3
Technologist	13	25	8	15.7
Total	52	100	51	100

Charité = Charité–Universitätsmedizin Berlin, UKE = Universitätsklinikum Eppendorf, Hamburg.

**Table 3 healthcare-10-02339-t003:** Countries of origin of survey participants at Charité (A) and UKE (B) corresponding to Figure 2.

A					
Charité—Country of Birth					
		Frequency	Percentage	Valid Percentage	Cumulative Percentage
Valid	Albania	7	4.9	5.8	5.8
	Belarus	1	0.7	0.8	6.6
	Bosnia and Herzegovina	1	0.7	0.8	7.4
	Brazil	2	1.4	1.7	9.1
	Bulgaria	4	2.8	3.3	12.4
	China	6	4.2	5.0	17.4
	Colombia	1	0.7	0.8	18.2
	Croatia	3	2.1	2.5	20.7
	Czech Republic	1	0.7	0.8	21.5
	Estonia	1	0.7	0.8	22.3
	Ethiopia	1	0.7	0.8	23.1
	France	2	1.4	1.7	24.8
	Ghana	1	0.7	0.8	25.6
	Greece	2	1.4	1.7	27.3
	Honduras	1	0.7	0.8	28.1
	Hungary	2	1.4	1.7	29.8
	India	2	1.4	1.7	31.4
	Indonesia	2	1.4	1.7	33.1
	Iran	3	2.1	2.5	35.5
	Iraq	1	0.7	0.8	36.4
	Israel	1	0.7	0.8	37.2
	Italy	8	5.6	6.6	43.8
	Kazakhstan	4	2.8	3.3	47.1
	Kenya	1	0.7	0.8	47.9
	Mexico	3	2.1	2.5	50.4
	Mongolia	1	0.7	0.8	51.2
	Netherlands	1	0.7	0.8	52.1
	Philippines	1	0.7	0.8	52.9
	Poland	8	5.6	6.6	59.5
	Portugal	2	1.4	1.7	61.2
	Romania	2	1.4	1.7	62.8
	Russia	4	2.8	3.3	66.1
	Saudi Arabia	3	2.1	2.5	68.6
	Serbia	2	1.4	1.7	70.2
	South Africa	1	0.7	0.8	71.1
	Spain	3	2.1	2.5	73.6
	Sri Lanka	2	1.4	1.7	75.2
	Suisse	1	0.7	0.8	76.0
	Sweden	1	0.7	0.8	76.9
	Syria	5	3.5	4.1	81.0
	Tajikistan	2	1.4	1.7	82.6
	Thailand	2	1.4	1.7	84.3
	Tunisia	1	0.7	0.8	85.1
	Turkey	7	4.9	5.8	90.9
	Ukraine	2	1.4	1.7	92.6
	USA	7	4.9	5.8	98.3
	Russia	1	0.7	0.8	99.2
	Vietnam	1	0.7	0.8	100.0
	Total	121	85.2	100.0	
NA		21	14.8		
Total		142	100.0		
**B**					
**UKE—Country of Birth**					
		Frequency	Percentage	Valid Percentage	Cumulative Percentage
Valid	Albania	1	0.9	1.0	1.0
	Argentina	1	0.9	1.0	2.1
	Austria	1	0.9	1.0	3.1
	Belarus	1	0.9	1.0	4.1
	Belgium	1	0.9	1.0	5.2
	Bosnia and Herzegovina	2	1.8	2.1	7.2
	Brazil	1	0.9	1.0	8.2
	Bulgaria	1	0.9	1.0	9.3
	Cameroon	1	0.9	1.0	10.3
	China	5	4.6	5.2	15.5
	Croatia	3	2.8	3.1	18.6
	Denmark	1	0.9	1.0	19.6
	Finland	1	0.9	1.0	20.6
	France	5	4.6	5.2	25.8
	Georgia	1	0.9	1.0	26.8
	Greece	1	0.9	1.0	27.8
	India	3	2.8	3.1	30.9
	Iran	9	8.3	9.3	40.2
	Ireland	1	0.9	1.0	41.2
	Israel	2	1.8	2.1	43.3
	Italy	3	2.8	3.1	46.4
	Japan	2	1.8	2.1	48.5
	Kazakhstan	1	0.9	1.0	49.5
	Kosovo	2	1.8	2.1	51.5
	Latvia	1	0.9	1.0	52.6
	Lebanon	1	0.9	1.0	53.6
	Luxembourg	1	0.9	1.0	54.6
	Morocco	1	0.9	1.0	55.7
	Nepal	1	0.9	1.0	56.7
	Netherlands	1	0.9	1.0	57.7
	Philippines	15	13.8	15.5	73.2
	Poland	5	4.6	5.2	78.4
	Portugal	3	2.8	3.1	81.4
	Romania	2	1.8	2.1	83.5
	Russia	1	0.9	1.0	84.5
	Saudi Arabia	2	1.8	2.1	86.6
	Serbia	2	1.8	2.1	88.7
	Somalia	1	0.9	1.0	89.7
	Spain	2	1.8	2.1	91.8
	Syria	3	2.8	3.1	94.8
	Tunesia	1	0.9	1.0	95.9
	United Arab Emirates	1	0.9	1.0	96.9
	USA	1	0.9	1.0	97.9
	Vietnam	2	1.8	2.1	100.0
	Total	97	89.0	100.0	
NA		12	11.0		
Total		109	100.0		

Abbreviaions: NA = not available, Charité = Charité–Universitätsmedizin Berlin, UKE = Universitätsklinikum Eppendorf, Hamburg.

**Table 4 healthcare-10-02339-t004:** Experiences of discrimination during the past six months at Charité. The proportion of staff with experiences of discrimination by peers differed significantly between the occupational groups at Charité (*p* = 0.006), with nurses and technicians reporting the most experiences of discrimination. We asked specifically about experiences of discrimination by supervisors, colleagues, and patients.

		Non-Scientific Staff			Clinical and Scientific Staff			Junior Staff			Other		
	Charité	Yes	No	Missing	Yes	No	Missing	Yes	No	Missing	Yes	No	Missing
		Number	Number	Number	Number	Number	Number	Number	Number	Number	Number	Number	Number
Experience of discrimination	by peers	16	34	2	8	37	1	2	34	0	0	4	1
	by superiors	8	39	5	9	35	2	4	30	2	2		0
	by patients	5	42	5	4	39	3	4	31	1		4	1

**Table 5 healthcare-10-02339-t005:** Experiences of discrimination based on migration background during the past 6 months at UKE. The proportion of staff with experiences of discrimination by peers differed significantly between the occupational groups at UKE (*p* = 0.002), with nurses and technicians reporting the most experiences of discrimination. We asked specifically about experiences of discrimination by supervisors, colleagues and patients based on the migration backgrounds of the participants.

		Non-Scientific Staff			Clinical and Scientific Staff			Junior Staff			Other		
	UKE	Yes	No	Missing	Yes	No	Missing	Yes	No	Missing	Yes	No	Missing
		Number	Number	Number	Number	Number	Number	Number	Number	Number	Number	Number	Number
Experience of discrimination based on the migration background	by peers	18	28	5	1	13	0	5	33	0	2	3	0
	by superiors	11	34	6	0	14	0	3	34	1	2	3	0
	by patients	9	31	11	2	8	4	0	4	34	0	5	0

**Table 6 healthcare-10-02339-t006:** Participants indicating being treated equally showed higher satisfaction with their working environment, as, for example, regarding the appreciation of work performance by colleagues (*p* < 0.001), superiors (*p* < 0.001) and patients (*p* = 0.030).

Perception of Equal Treatment	Charité		UKE	
	N	%	N	%
Contract	138	100	102	100
Yes	112	81.2	84	82.4
No	26	18.8	18	17.6
Weekly working hours	140	100	104	100
Yes	127	90.7	89	85.6
No	13	9.3	15	14.4
Working times	120	100	95	100
Yes	101	84.2	82	86.3
No	19	15.8	13	13.7
Payment	136	100	100	100
Yes	108	79.4	82	82
No	28	20.6	18	18
Distribution of tasks within team	137	100	100	100
Yes	107	78.1	75	75
No	30	21.9	25	25
Appreciation of work performance by colleagues	133	100	100	100
Yes	109	82	74	74
No	24	18	26	26
Appreciation of work performance by superiors	132	100	103	100
Yes	99	75	77	74.8
No	33	25	26	25.2
Appreciation of work performance by patients	99	100	63	100
Yes	89	89.9	47	74.6
No	10	10.1	16	25.4
Opportunities for further training and promotion	129	100	94	100
Yes	79	61.2	63	67
No	50	38.8	31	33
Opportunity to express ones opinion in discussions among colleagues	135	100	96	100
Yes	108	80	77	80.2
No	27	20	19	19.8

## Data Availability

The data presented in this study are available on request from the corresponding author.

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
