# Peer review of "Foreign Healthcare Professionals in Germany: A Questionnaire Survey Evaluating Discrimination Experiences and Equal Treatment at Two Large University Hospitals"

_healthcare, 2022, doi:10.3390/healthcare10122339_

Round 1
Author Response
Dear Reviewer 1
Thank you very much for your time and thorough review of our submitted manuscript. Your comments really made us think and have helped us tremendously in improving our manuscript. With your input, we have extensively revised the presentation of our study and hope that the manuscript is now deemed suitable for the international readership of your journal.
Please find below our point-by-point responses to your criticism. We have consecutively numbered the issues raised including our responses and provide the numbers in the revised manuscript (using the comment function) for straightforward identification of all changes made in response to your criticism.

Reviewer 2 Report
Dear Authors!
The theme is interesting and always current. Even so, the study needs improvement, namely in terms of methodology, discussion of results and conclusion of the work.
The "Study design" should identify the type of study from the methodological point of view and the reasons for the choice.
In the "Questionnaire" it would be interesting to have an access or attached information more detailed about the algorithm of the questionnaire.
In the discussion when it is stated "In this respect, the linking of racism research with critical migration research is obvious." The data presented is not categorical, and that a strong relationship between the two is evident is the correct thing to say.
The discussion has nonsensical paragraphs that cut the connection and meaning of the content. It should be revised.
The discussion is also a bit confused and the connection between the results and the confrontation with the available evidence is not always clearly explained. As a suggestion would be to divide the discussion into subchapters.
Some citations are very old, if possible update the references.
The conclusion with the summary of the findings and their relevance is missing.
with best regards
Author Response
Dear Reviewer 2
Thank you very much for your review and valuable suggestions to improve our manuscript. We hope that our revisions meets your approval.
Please find below our point-by-point responses to your criticism. We have consecutively numbered the issues raised including our responses and provide the numbers in the revised manuscript (using the comment function) for straightforward identification of all changes made in response to your criticism.

Reviewer 3 Report
I have reviewed the paper titled 'Foreign healthcare professionals in Germany: A questionnaire survey evaluating work satisfaction and discrimination experiences at two large university hospitals'.
In the main the paper reads well, but I have made some edits and corrections in sticky notes added to the PDF of the original submission - attached herewith as the supporting documentation of my review.
The Methods and Results sections need to be improved, as I picked up a number of discrepancies in the findings - please review my comments in the sticky notes in terms of discrepancies in the findings and how they have been presented and please provide clarification of the discrepancies (i.e. explain comprehensively in the text and graphics) or make the necessary corrections. The tables need attention in terms of standardising their formats and some extra information is required in the column titles, as well as the numbers reflected in the main body of the table - please see my comments in the sticky notes.
The References section also needs attention - the font type and size need to be the same for all the references, and some URL links do not work - please insert the updated URL links and revised dates of access.

Author Response
Dear Reviewer 3
We are very grateful for your valuable time and thorough and detailed review of our manuscript. Your critical comments have helped us very much in improving our manuscript.
Please find attached our point-by-point responses to your criticism. We have consecutively numbered the issues raised including our responses and provide the numbers in the revised manuscript (using the comment function) for straightforward identification of all changes made in response to your criticism.

Round 2
Reviewer 2 Report
Dear Authors!
I think that the changes made have responded to the suggestions I made.
With best regards